# Nutrient Adequacy Is Low among Both Self-Declared Lacto-Vegetarian and Non-Vegetarian Pregnant Women in Uttar Pradesh

**DOI:** 10.3390/nu12072126

**Published:** 2020-07-17

**Authors:** Alexandra L. Bellows, Shivani Kachwaha, Sebanti Ghosh, Kristen Kappos, Jessica Escobar-Alegria, Purnima Menon, Phuong H. Nguyen

**Affiliations:** 1Department of International Health, Johns Hopkins Bloomberg School of Public Health, Johns Hopkins University, Baltimore, MD 21205, USA; alexandra.bellows@jhu.edu; 2International Food Policy Research Institute, Washington, DC 20005, USA; S.Kachwaha@cgiar.org (S.K.); P.Menon@cgiar.org (P.M.); 3FHI Solutions, Washington, DC 20001, USA; sghosh@fhi360.org (S.G.); KKappos@fhi360.org (K.K.); JEscobar-alegria@fhi360.org (J.E.-A.)

**Keywords:** India, dietary intake, dietary diversity, vegetarian, maternal nutrition, pregnancy

## Abstract

Poor dietary intake during pregnancy remains a significant public health concern, affecting the health of the mother and fetus. This study examines the adequacy of energy, macronutrient, and micronutrient intakes among self-declared lacto-vegetarian and non-vegetarian pregnant women. We analyzed dietary data from 627 pregnant women in Uttar Pradesh, India, using a multiple-pass 24 h diet recall. Compared to non-vegetarians, lacto-vegetarians (~46%) were less likely to report excessive carbohydrate (78% vs. 63%) and inadequate fat intakes (70% vs. 52%). In unadjusted analyses, lacto-vegetarians had a slightly higher mean PA for micronutrients (20% vs. 17%), but these differences were no longer significant after controlling for caste, education, and other demographic characteristics. In both groups, the median intake of 9 out of 11 micronutrients was below the Estimated Average Requirement. In conclusion, the energy and micronutrient intakes were inadequate, and the macronutrient intakes were imbalanced, regardless of stated dietary preferences. Since diets are poor across the board, a range of policies and interventions that address the household food environment, nutrition counseling, behavior change, and supplementation are needed in order to achieve adequate nutrient intake for pregnant women in this population.

## 1. Introduction

Undernutrition and micronutrient deficiencies in pregnant women are significant public health concerns in many low-and middle-income countries (LMICs) [1,2,3,4]. It is estimated that approximately 822 million people (11% of the global population) are undernourished—not having enough food to meet energy requirements [5]. In addition, 30% of women of reproductive age and 40% of pregnant women worldwide suffer from anemia [6]. Maternal undernutrition prior to and during pregnancy can lead to adverse outcomes, including fetal growth restriction, the risk of preterm birth, and maternal mortality [3,7]. Fetal growth restriction also increases a child’s risk of stunting and poor cognitive development [3,8,9]. Deficiencies in key micronutrients such as calcium, iron, and folate are also associated with the increased risk of hypertension, maternal death, low birthweight of the child, and neural tube defects [3,10,11,12,13].

Immediate causes for undernutrition and micronutrient deficiencies are frequent illness and the lack of adequate nutritious foods [14]. A systematic review of dietary intake during pregnancy in LMICs highlighted that diets were primarily reliant on cereal grains, with excessive intakes of carbohydrates and insufficient intakes of micronutrients [1]. Specifically, pregnant women had a low intake of energy, protein, fat, folate, iron, vitamin A, vitamin C, zinc, and calcium [1]. 

The typical determinants of diet quality include socioeconomic status (SES), food security, maternal knowledge of nutrition, maternal education, food preferences, and external food environment [15,16,17,18,19]. Limited research has been conducted in LMICs to assess how dietary preferences, including preferences for vegetarian diets, may affect diet quality during pregnancy. Due to the purposeful avoidance of many animal-source foods, vegetarian women may be at increased risk for micronutrient deficiencies and insufficient protein intake. However, a systematic review on vegetarianism during pregnancy found mixed results and concluded vegetarian diets to be safe if the micronutrient intake was adequate [20,21]. The majority of studies in this review were from high-income countries, and less is known, largely due to limited dietary intake data availability, from low- and middle-income settings, such as India [22].

In India, the National Family Health Survey (2015–2016) found that approximately 30% of women of reproductive age practice vegetarianism [23], and the practice is associated with cultural and religious traditions. However, to our knowledge, no study has assessed the differences in macro and micronutrient intake of self-identified lacto-vegetarian and non-vegetarian pregnant women in India. Such an assessment is valuable, because discussions about the role of dietary preferences in determining dietary intakes should be based on high-quality dietary data.

The goals of this study are, therefore, to (1) examine the adequacy of energy, macronutrient, and micronutrient intakes among lacto-vegetarian and non-vegetarian pregnant women in Uttar Pradesh, India; and (2) assess the contribution of different food groups to energy, macronutrient, and micronutrient intakes in this population.

## 2. Materials and Methods

### 2.1. Data and Study Population

We used data collected at baseline from the impact evaluation of a maternal nutrition intervention implemented by Alive and Thrive (A&T) in Uttar Pradesh (ClinicalTrials.gov Identifier: NCT03378141). A&T’s intervention aims to address maternal undernutrition in Uttar Pradesh by testing the feasibility of improving the provision and uptake of a package of maternal nutrition interventions in existing antenatal services [24]. The results of the impact evaluation will be reported in late 2020. The baseline data were collected by independent evaluators from International Food Policy Research Institute (IFPRI) and Network for Engineering and Economics Research and Management (NEERMAN) between October 2017 and December 2017. Details of the baseline household survey have been described elsewhere [24,25]. Briefly, two districts in Uttar Pradesh were selected to be part of the nutrition intervention evaluation. Within these districts, 26 blocks were selected, and within each block 7 Gram Panchayats were identified. For each Gram Panchayat, we used a systematic random sampling methodology to select four households with a pregnant woman in the second or third trimester of pregnancy who was eligible for participation, yielding a total sample size of 655 eligible pregnant women. This study was approved by the ethical review board (IRB) from Suraksha Independent Ethics Committee in India IRB (approval number: 2017-10-9094) and the International Food Policy Research Institute (IRB approval number: 17-09-11). Verbal informed consent was obtained from all the pregnant women.

### 2.2. Dependent Variables

Dietary intake was assessed using a single multiple-pass 24 h diet recall conducted by a trained enumerator. The women were asked to recall which foods and beverages they consumed in the preceding 24 h period. They were asked to name the foods or mixed dishes they consumed, the time of day they consumed each food or beverage item, and quantity of the food or beverage. For mixed dishes, the women were asked to list the individual ingredients and describe the recipe for each dish. To help the women determine quantity, standard visual aids were presented at the time of interview.

Foods were categorized into food groups based on the Minimum Dietary Diversity for Women guidelines [26], which consist of the following 10 food groups: (1) starchy staple foods; (2) beans and peas; (3) nuts and seeds; (4) dairy products; (5) meat, poultry, and fish; (6) eggs; (7) dark green leafy vegetables; (8) vitamin A-rich fruits and vegetables; (9) other vegetables; and (10) other fruits. The dietary diversity score was calculated as the sum of the 10 food group categories. The minimum dietary diversity for women (MDD-W) was defined as consuming five or more food groups (out of ten) within the last 24 h [26]. A score of five or more has been associated with an increased likelihood of nutritional adequacy in non-pregnant non-lactating women of reproductive age [27]. We created an additional food group for unhealthy foods, which consisted of sugar-sweetened beverages, candies, chocolate, frozen treats, cakes, pastries, sweet biscuits, chips, crisps, and other foods high in sodium and sugar. The unhealthy food category was not included in the calculation for the minimum dietary diversity score.

We estimated the macronutrients (energy, carbohydrates, protein, and fat) and 11 micronutrients (iron, calcium, zinc, vitamin A, thiamin, riboflavin, niacin, pyridoxine, cobalamin, vitamin C, and folate) using the 2017 Indian Food Composition Table [28] and appropriate conversion factors to take into account the loss of nutrients from cooking methods. We calculated the estimated energy requirement (EER) for women ≥ 19 years using the following formula: EER = 354 − (6.91 × age [y]) + physical activity × [(9.36 × weight [kg]) + (726 × height [m]) [29]. For women enrolled in our study who were 18 years of age, we calculated the EER using the following formula: EER = 135.3 − (30.8 × age [year]) + physical activity × [(10.0 × weight [kg]) + (934 × height [m]) +25 [29]. For the physical activity coefficient, we estimated this value based on participant’s occupation and age (farmers age 19+ = 1.27, farmers age 18 = 1.31, other occupation age 19+ = 1.12, other occupation age 18 = 1.16). Data on the pre-pregnancy BMI was not available, therefore we used estimates for increased energy requirements necessary for appropriate weight gain for a woman with a normal BMI pre-pregnancy. We added an additional energy requirement for pregnancy of 350 kcal/d for the second trimester and 500 kcal/d for the third trimester of pregnancy [30]. We used the Goldberg method [31] to identify the extreme outliers of under and over reporting participants. We excluded 28 women from the analysis because their reported energy intake fell outside of the 95% confidence interval for the Goldberg ratio (22 under reporters and 6 over reporters). We also examined the macronutrient intake between lacto-vegetarian and non-vegetarian participants using the Acceptable Macronutrient Distribution Ranges (AMDR) to determine insufficient, optimal, and excessive intakes of carbohydrate, fat, and protein. The optimal ranges for carbohydrate, fat, and protein were categorized as 45–65%, 20–35%, and 10–35% of total energy, respectively [29].

To obtain the usual intake distributions for each micronutrient, we used a measurement-error model taking into account within-person variation in observed nutrient intake [32]. We applied a Box-Cox transformation to obtain a normal distribution for each nutrient and calculated the best linear unbiased predictor for each nutrient [33]. Since only one recall was obtained from our study population, we used the standard errors corrected for within-person variation obtained from repeated 24 h recalls from a similar population of pregnant women in Uttar Pradesh [34].

We calculated the probability of adequacy as the probability of an individual’s usual intake being above the Estimated Average Requirement (EAR) for each micronutrient [35]. We also adjusted the EAR for iron to account for 10% bioavailability [35] and zinc to account for diets high in phytates [36]. For calcium, we used the Adequate Intake due to a lack of EAR available [37]. We calculated the mean probability of adequacy (MPA) as the mean probability of adequacy for the selected 11 micronutrients.

### 2.3. Independent Variable

To assess dietary preferences, we asked women if they were usually vegetarian, with the response options: “no”, “yes, all the time”, or “yes, sometimes”. For the purposes of this analysis, we considered women to be vegetarian only if they responded “yes, all the time”; otherwise, we categorized these women to be non-vegetarian. Vegetarian in this population was defined as lacto-vegetarian, abstaining from eating meat and eggs but consuming dairy.

### 2.4. Other Variables

At the individual level, we collected information on women’s age, height, weight, religion, caste, education, occupation, number of previous pregnancies, and current gestational age. At the household level, we collected information on household food insecurity and SES. The household food insecurity index was constructed using the Household Food Insecurity Access Scale [38]. This tool consists of nine questions related to the household’s experience with food insecurity in the past 30 days [38]. We categorized participants using the United States Agency for International Development’s Food and Nutrition Technical Assistance III Project (FANTA) criteria [38] as either food-secure or food-insecure (mild, moderate, or severe). We constructed the household SES index using a factor analysis, which include household ownership of selected assets, land ownership, and housing characteristics [39]. At the community level, we collected information regarding perceived social norms of dietary practices during pregnancy. These questions focused on behaviors such as dietary diversity, quantity of food, and food avoidance practices. The responses to these questions were measured on a 5-point Likert scale by asking women the extent to which they felt their community agreed or disagreed with certain maternal nutrition practices.

### 2.5. Statistical Analysis

We compared demographic characteristics and differences in dietary intake for lacto-vegetarian and non-vegetarian pregnant women using chi-square tests for categorical variables and the Student’s t-test for continuous variables. We used a multivariable regression model to assess the differences in macronutrient and micronutrient intake between lacto-vegetarian and non-vegetarian women, controlling for food security (food secure or food insecure), caste classification (general, Scheduled Caste/Scheduled Tribe, Other Backward Classes), maternal education (no schooling, primary school, or secondary school and above), parity (continuous), and SES (tertiles). Since the MPA was not normally distributed, we used the square-root transformation of MPA as the dependent variable in the regression model. Additionally, for MPA we ran a second model that controlled for the covariates previously mentioned and the natural log transformed energy intake. All the models controlled for clustering by including a random intercept for Gram Panchayats. All the analyses were conducted using STATA 15 IC (STATA Corp).

## 3. Results

### 3.1. Demographic Characteristics

A total of 627 pregnant women were included in our analysis. The prevalence of lacto-vegetarianism was 46.4%. The mean age was 25.0 years (SD: 4.1; range: 18 to 40), with no differences between lacto-vegetarian and non-vegetarian women. Lacto-vegetarians were more likely to identify their religious beliefs as Hindu (92.7% vs. 88.1%) and attend high school (50.5% vs. 28.9%) compared to non-vegetarians. Lacto-vegetarian women were more likely to come from households with a higher SES, to be from privileged caste communities, and were less likely to report food insecurity (Table 1). We found no significant differences in the perceived social norms of dietary practices during pregnancy between the two groups (Figure 1).

### 3.2. Dietary Patterns and Diet Diversity

Pregnant women reported consuming an average of 4.2 (SD: 1.1) food groups out of 10. Diets were poor overall, and there were no differences in the average number of food groups consumed between lacto-vegetarian and non-vegetarian women (4.2 vs. 4.1; *p*-value > 0.05). Among non-vegetarians, only 8.0% consumed flesh foods and 4.0% consumed eggs. There were no statistically significant differences in the consumption of food groups besides eggs and flesh floods between the lacto-vegetarian and non-vegetarian women (Figure 2). The lacto-vegetarian women were more likely to meet the MDD-W compared to the non-vegetarian women (39.8% compared to 34.5%, respectively), but the differences between the groups were not statistically significant (*p* = 0.52). The unhealthy food consumption (e.g., salty and sweet snack foods) was also higher in the lacto-vegetarian (40.9%) than the non-vegetarian women (30.4%), but an adjusted analysis found no statistically significant difference between the groups (*p* > 0.05).

### 3.3. Energy and Macronutrient Intake

The lacto-vegetarian women reported a slightly higher energy intake compared to the non-vegetarian women (2052 vs. 1949 kcal/day) (Table 2). In both groups, the women consumed on average less than their EER. Significantly more women in the non-vegetarian group reported an excessive intake of carbohydrates (78.7%) compared to the lacto-vegetarians (63.5%). In addition, the majority of women in both groups reported an insufficient intake of fat, with a significantly higher percentage of women in the non-vegetarian group (Table 2). No significant difference was found in the protein intake between the two groups, with an average protein intake at 11.8% of the total energy and 89.0% of women consuming the optimal amount of protein (Table 2).

### 3.4. Micronutrient Intake

Both lacto-vegetarian and non-vegetarian groups reported an average intake less than the EAR for 9 out of the 11 micronutrients (Table 3). Zinc and thiamin were the only micronutrients where the average intake was above the EAR for both groups. The lacto-vegetarian women had a significantly higher probability of adequacy for calcium and vitamin B12 compared to the non-vegetarian women. In the unadjusted model, the lacto-vegetarian women had a slightly higher MPA compared to the non-vegetarian women (19.9% vs. 17.2%; *p* = 0.02), but these differences were no longer significant after controlling for energy intake, food security status, caste affiliation, maternal education, SES, and parity (Table 4).

### 3.5. Contribution of Food Groups to Macro- and Micronutrients

For both groups, the majority of energy came from starchy staple foods, which accounted for 62.4% of the total energy for lacto-vegetarians and 69.1% of the total energy for non-vegetarians. Starchy staples were also the major dietary source of carbohydrates, protein, iron, folate, and zinc for both groups. Dairy products constituted 48.6% of the calcium intake and 99.6% of the vitamin B12 intake for lacto-vegetarians and 36.8% of the calcium intake and 89.6% of the vitamin B12 intake for non-vegetarians. Fats and oils, added sugar, and unhealthy foods contributed to 13.2% of energy for non-vegetarians and 16.6% of energy for lacto-vegetarians, yet contributed minimally to the micronutrient intake (Figure 3).

## 4. Discussion

We found that the diets of pregnant women in Uttar Pradesh are poor, regardless of their own stated dietary preferences. The energy intakes were inadequate and the macronutrient intakes were imbalanced for both the self-declared lacto-vegetarian and non-vegetarian women. The micronutrient adequacy was low (18%), regardless of dietary preferences. For both groups, the mean micronutrient intake was less than the EAR for 9 out of the 11 micronutrients assessed. The difference in mean micronutrient adequacy between the two groups was no longer significant after controlling for food security, caste, maternal education, parity, SES, and energy intake. In the final model, only SES had a significant association with micronutrient adequacy.

We only found one other small study (*n* = 49 participants) that assessed the differences in dietary intake of vegetarian and non-vegetarian pregnant women in Pune, India [40], which found that vegetarian women had lower levels of serum folic acid and vitamin B12 compared to non-vegetarian women. These findings are in contrast to ours, which found that the lacto-vegetarian women had higher intakes of vitamin B12, most likely due to their higher intake of dairy products and the non-vegetarians’ low intake of other animal-source foods (<8%). These differences could be due to the poor correlation between dietary intake and serum biomarkers for folate and vitamin B12 [41] or differences in the assessment of vegetarian status. The study in Pune assessed vegetarian status based on dietary patterns instead of self-identification, therefore it may not have been able to distinguish differences in intake due to dietary preferences versus economic restrictions. Our analysis suggests a strong intersection of these, as expected.

Our results on the poor quality of diets are consistent with other studies assessing the dietary intake of women of reproductive age in India and South Asia. A study conducted outside of Delhi found that pregnant women in rural areas had insufficient intakes of energy, protein, iron, folic acid, and vitamin C [42]. In Nepal, lactating women had a mean MPA of 0.19, which was similar to our findings [43]. In contrast, Bangladeshi pregnant women had a higher energy intake (2311 vs. 2051 kg/day), dietary diversity score (5.1 vs. 4.1 food groups), and MPA (0.40 vs. 0.18) compared to the women in our study [33]. Given the similar resource constraints and income levels in both contexts, the better dietary intakes in Bangladesh (~80% consumed animal source foods) could be due to higher nutrition knowledge, more supportive social norms, and the accessibility and affordability of animal-source foods.

Interestingly, we found that greater than one third of pregnant women consumed unhealthy foods. The unhealthy foods most commonly reported were namkeen (a savory Indian snack) and biscuits without cream. While an unadjusted analysis found that lacto-vegetarian women were more likely to consume unhealthy foods compared to non-vegetarians, these differences were no longer significant in an analysis that adjusted for SES. Therefore, here again, the SES differences between lacto-vegetarians and non-vegetarians may account for differences in consumption.

The women in our study consumed below the recommended energy intake, excessive carbohydrates, insufficient fat, and had a low probability of micronutrient adequacy, therefore they may be at risk for inadequate weight gain and adverse pregnancy outcomes [44]. Inadequate dietary intakes during pregnancy could be the result of food insecurity, food habits, low SES, external food environment, or individual and community beliefs [17]. In our study, approximately 15% of households reported some level of food insecurity and 46% practiced lacto-vegetarianism, much higher than the national average of 30% [23]. A recent study assessing the cost of nutritious diets in the same context found that inadequate dietary intakes were associated with food habits and unaffordability due to income levels rather than food availability [45]. Social norms unfavorable to adequate diets were also prevalent in this population, with 20% of women stating that people in their community felt women should not eat too much to avoid a difficult labor. A study in Tamil Nadu, India, found that pregnant women avoided “hot foods”, including meat and eggs [46]. If similar food beliefs are present in this community, this could potentially explain the low consumption of meat and eggs even among the non-vegetarian pregnant women.

Dietary intakes are influenced by complex interactions of individual, household, community, and environmental factors. An explanation for the unadjusted results showing that self-declared lacto-vegetarian women had a higher probability of adequate intake of micronutrients could be a more favorable SES, higher education, less food insecurity, and differences in the caste composition of the two groups. Indeed, after controlling for energy intake, caste, demographic, and SES factors, we found no significant difference in the micronutrient adequacy between the lacto-vegetarian and non-vegetarian women. The non-vegetarian women reported very low intakes of meat, poultry, and fish, which are excellent sources of iron, zinc, vitamin A, and vitamin B12. Animal source foods and other food sources rich in micronutrients are typically more expensive than cereals and grains [47]. The consumption of these foods is usually dependent on SES and food insecurity.

This study provides unique insights into the dietary patterns and nutrient intakes among lacto-vegetarian and non-vegetarian pregnant women in India, a topic which has been largely unexplored in the literature until now. Few studies have looked at dietary intake of pregnant women in India, but no study to our knowledge has assessed the differences in dietary intake between self-identified lacto-vegetarians and non-vegetarian pregnant women in an LMIC setting. This study benefited from the use of a 24 h recall, allowing us to estimate the probability of nutrient adequacy. The limitations of this study include the cross-sectional study design, which prevents us from assessing causality, and the reliance on one 24 h recall to assess dietary intake. The dietary results may have inflated standard errors due to within-person variation and may be influenced by seasonality. To account for within-person variation, we used corrected standard errors from an external study, which collected two 24 h recalls on a subsample of the population. Additional concerns may be related to social desirability and recall bias. The lacto-vegetarian women were more educated, which may suggest an advantage on their ability to recall [48]. We used a multiple pass method which involved repeating list of foods consumed and asking probing questions using visual aids in order to improve recall in both groups. Finally, social desirability may lead to the under reporting of animal source food consumption if non-vegetarians are concerned about potential social discrimination in communities where socially and economically dominant groups promote and prefer vegetarian diets.

Our results indicate that diets during pregnancy are poor for both lacto-vegetarian and non-vegetarian women. A range of policies and interventions that address the food environment, nutrition counselling, behavior change, and supplementation are needed in order to achieve adequate nutrient intake for pregnant women in this population. For interventions that promote nutrition counselling, practitioners should be aware that even non-vegetarian women may be unable to consume adequate amounts of animal source foods due to financial constraints. Behavior change interventions that increase maternal nutritional knowledge and contribute to adopting the recommended practices should be combined with social protection strategies to ensure that micronutrient-rich foods are affordable to all members of the population. In contexts such as India, protein-energy supplements, multiple micronutrient supplementation, and the fortification of existing food transfer programs may also be useful as short-term strategies to achieve recommended energy and nutrient adequacy [49].

## Figures and Tables

**Figure 1 nutrients-12-02126-f001:**
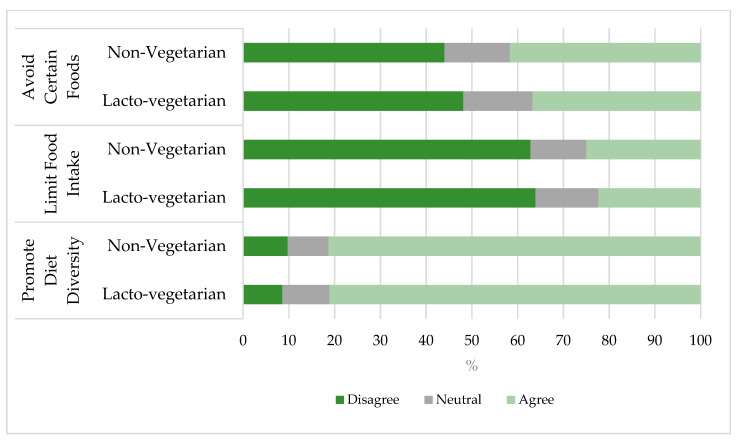
Social norms about diet and nutrition during pregnancy. Participants were asked the following questions regarding social norms in their community. Promote Diet Diversity: in my family and community we/people expect pregnant women to consume five varieties and larger quantity of food to get enough energy and nutrition during pregnancy. Limit Food Intake: most people who are important to me (e.g., family members, friends…) think that a pregnant woman should not eat too much to avoid difficult labor due to large baby. Avoid Certain Foods: in my family and community, pregnant women are expected to avoid certain kinds of foods (like meat, fish, papaya, jackfruit, milk etc.) because it will harm the mother and/or baby.

**Figure 2 nutrients-12-02126-f002:**
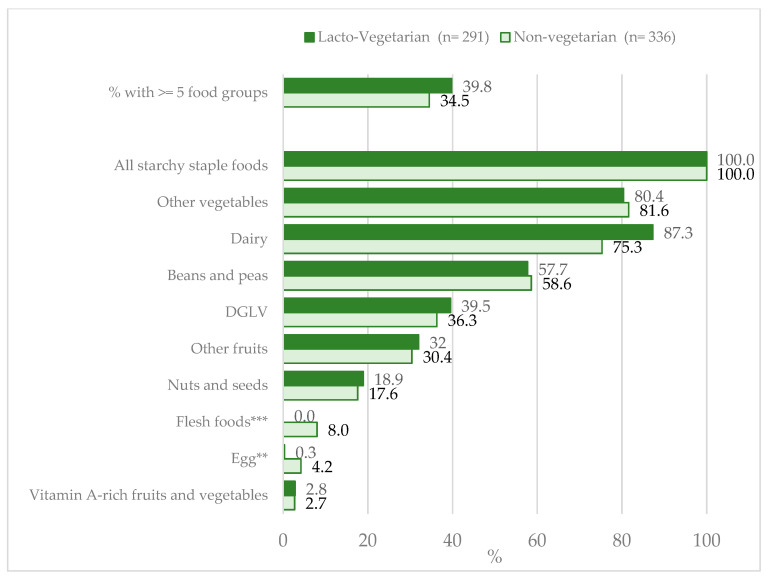
Food group consumption in the previous 24 h among lacto-vegetarian and non-vegetarian pregnant women. DGLV: Dark Leafy Green Vegetables; flesh foods: meat, poultry, and fish. ** *p* < 0.01, *** *p* < 0.001.

**Figure 3 nutrients-12-02126-f003:**
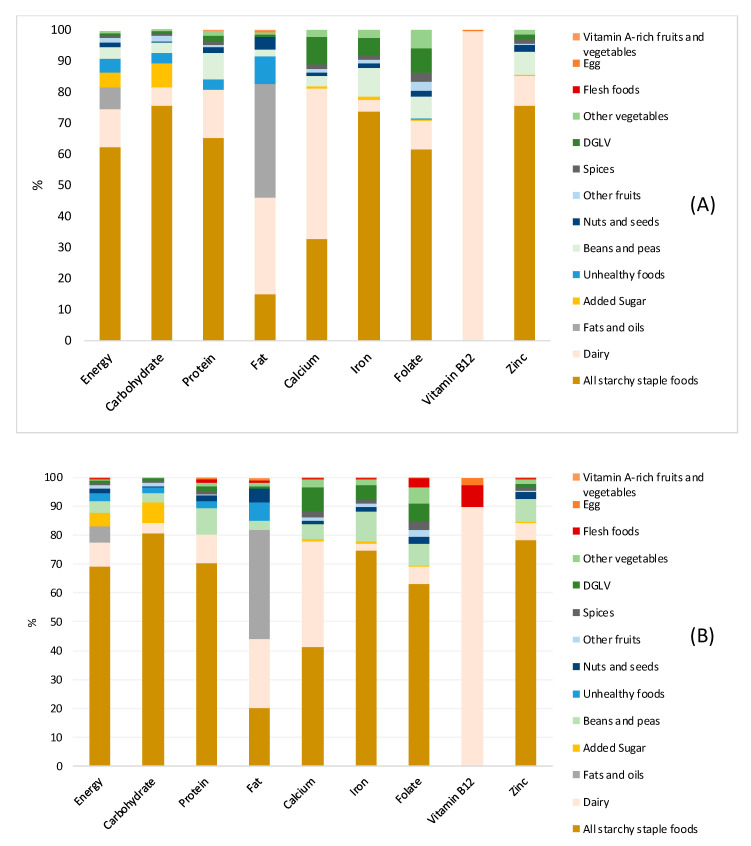
(**A**). Lacto-vegetarians, (**B**). Non-vegetarians. Relative contributions of different food groups to the dietary intakes of selected micronutrients among lacto-vegetarian and non-vegetarian pregnant women in Uttar Pradesh, India.

**Table 1 nutrients-12-02126-t001:** Descriptive statistics of the study participants ^1^.

	All	Lacto-Vegetarian	Non-Vegetarian	*p*-Values
	*N* = 627	*N* = 291	*N* = 336	
Participant characteristics				
Maternal age, year	25.0 ± 4.1	25.0 ± 3.9	25.0 ± 4.3	0.90
Religion as Hindus, %	92.7	97.9	88.1	<0.001
Caste category, %				
SC/ST	42.1	26.5	55.7	<0.001
OBC	40.8	50.2	32.8	
General	17.1	23.4	11.6	
Education, %				
No schooling	25.0	17.2	31.9	<0.001
Elementary school	14.8	11.0	18.2	
Middle school	21.2	21.3	21.1	
≥High school	38.9	50.5	28.9	
Occupation as housewives, %	90.8	91.4	90.7	0.78
Number of previous pregnancies, n	2.1 ± 1.9	1.9 ± 1.8	2.3 ± 2.0	0.003
Current gestational age in trimester, %				
Second trimester	43.2	40.2	45.8	0.16
Third trimester	56.8	59.8	54.2	
Household factors				
Number of people in household, n	5.1 ± 2.1	5.2 ± 2.0	5.1 ± 2.1	0.33
Household food insecurity, %	15.9	10.7	20.5	0.001
Household SES index	3.0 ± 1.4	3.3 ± 1.4	2.7 ± 1.4	<0.001

^1^ Values are mean ± SD or percentage. OBC: other backward classes; SC: scheduled caste; SES: socio-economic status; ST: Scheduled tribe.

**Table 2 nutrients-12-02126-t002:** Energy and macronutrient consumption among lacto-vegetarian and non-vegetarian pregnant women in Uttar Pradesh, India ^1^.

	Lacto-Vegetarian	Non-Vegetarian	Unadjusted *p*-Values	Adjusted *p*-Values ^c^
	*N* = 291	*N* = 336		
**Energy, kcal/day**	2051.5 ± 724.6	1949.0 ± 730.4	0.08	0.78
EER ^a^, kcal/day	2254.4 ± 147.1	2221.2 ± 143.1	0.004	0.31
Energy intake < 85% of EER, %	45.7	53.3	0.06	0.34
**Carbohydrate**				
Amount consumed, g (SD)	346.7 ± 124.7	346.4 ± 137.3	0.98	0.48
% energy	68.0 ± 8.5	71.1 ± 8.3	<0.001	0.003
Percent insufficient intake ^b^	1.0	0.9	0.90	0.82
Percent in optimal range	35.4	20.8	<0.001	0.01
Percent excessive intake	63.5	78.3	<0.001	0.009
**Fat**				
Amount consume, g	45.9 ± 27.2	35.1 ± 23.4	<0.001	0.004
% energy	19.8 ± 8.6	16.2 ± 8.2	<0.001	0.001
Percent insufficient intake ^b^	51.9	70.2	<0.001	0.004
Percent in optimal range	43.6	26.8	<0.001	0.001
Percent excessive intake	4.5	3.0	0.32	0.87
**Protein**				
Amount consume, g	59.9 ± 22.3	57.3 ± 22.2	0.15	0.82
% energy	11.7 ± 1.5	11.8 ± 1.6	0.40	0.78
Percent insufficient intake ^b^	11.7	10.4	0.60	0.62
Percent in optimal range	88.3	89.6	0.60	0.62
Percent excessive intake	0.0	0.0	1.0	1.0

^1^ Values are means ± SDs or percentages. ^a^ Estimated energy requirement (EER) for women >= 19 year was calculated based on the formula suggested by IOM: EER = 354 − (6.91 × age [year]) + PA × [(9.36 × weight [kg]) + (726 × height [m])], where PA (physical activity coefficient) = 1.27 for farmers and 1.12 for other occupations. EER for women aged 18 was calculated based on the formula suggested by IOM for 14–18 year olds. EER = 135.3 − (30.8 × age [year]) + physical activity × [(10.0 × weight [kg]) + (934 × height [m]) +25 [29]. Due to the increased energy needs during pregnancy, EER added 350 kcal/d for PW in the 2nd trimester and 500 kcal/day for PW in the 3rd trimester. ^b^ Insufficient and excessive intake levels were defined based on the following Acceptable Macronutrient Distribution Ranges (AMDRs): protein, 10–35%; fat, 20–35%; carbohydrate, 45–65% of total energy. ^c^ Adjusted *p*-values controlled for food security status, caste, maternal education, parity, and SES status.

**Table 3 nutrients-12-02126-t003:** Dietary intakes of selected micronutrients among lacto-vegetarian and non-vegetarian pregnant women in Uttar Pradesh, India.

	EAR (Mean ± SD)	Median (IRQ)				Probability of Adequacy (Mean ± SD)			
		Lacto-Vegetarian	Non-Vegetarian	Unadjusted *p*-Values	Adjusted *p*-Value **	Lacto-Vegetarian	Non-Vegetarian	Unadjusted *p*-Values	Adjusted *p*-Value **
		*N* = 291	*N* = 336			*N* = 291	*N* = 336		
MPA						19.9 ± 15.3	17.2 ± 13.7		
Calcium, mg *	800.0 ± 100	414.6 (232.1, 727.2)	270.7 (174.1, 522.6)	<0.001	<0.001	31.5 ± 26.6	20.1 ± 21.2	<0.001	<0.001
Iron, mg	24.9 ± 2.3	12.9 (9.2, 16.9)	12.5 (8.7, 17.1)	0.14	0.27	4.3 ±16.9	2.4 ± 11.8	0.10	0.28
Zinc, mg	8.0 ± 1.0	8.4 (6.2, 11.1)	8.4 (5.7, 10.7)	0.18	0.51	51.9 ± 38.2	50.0 ± 39.5	0.53	0.92
Vitamin C, mg	70.0 ± 7.0	40.2 (24.3, 61.5)	33.7 (18.8, 56.4)	0.11	0.54	15.4 ± 33.9	13.0 ± 32.0	0.38	0.90
Thiamin, mg	1.2 ± 0.1	1.3 (1.0, 1.7)	1.2 (0.9, 1.7)	0.11	0.38	57.6 ± 42.2	51.9 ± 43.3	0.10	0.33
Riboflavin, mg	1.2 ± 0.1	0.8 (0.6, 1.2)	0.7 (0.5, 1.0)	0.02	0.49	18.5 ± 34.7	12.1 ± 29.1	0.01	0.27
Niacin, mg	14.0 ± 2.1	12.7 (9.3, 16.1)	12.3 (9.1, 17.1)	0.73	0.61	35.8 ± 36.5	37.0 ± 38.1	0.70	0.58
Vitamin B6, mg	1.6 ± 0.2	0.7 (0.4, 1.0)	0.6 (0.4, 0.8)	0.003	0.17	1.0 ± 7.3	0.9 ± 7.7	0.85	0.84
Folate, mcg	520.0 ± 52.0	211.3 (154.8, 283.5)	203.3 (143.6, 280.5)	0.60	0.91	0.4 ± 5.8	1.1 ± 8.9	0.21	0.15
Vitamin B12, mcg	2.2 ± 0.2	0.6 (0.2, 1.4)	0.3 (0.1, 1.0)	<0.001	0.04	2.4 ± 11.9	0.2 ± 2.0	<0.001	0.02
Vitamin A, mcg RAE	550.0 ± 55.0	31.8 (17.6, 103.1)	32.4 (15.3, 73.5)	0.44	0.60	0.1 ± 0.9	0.1 ± 1.2	0.86	0.60

MPA: mean probability of adequacy; RAE: retinol activity equivalents. * Adequate intake, because there is no established EAR. ** Adjusted *p*-value controls for food security status, caste, maternal education, parity, and SES status.

**Table 4 nutrients-12-02126-t004:** Multivariable regression of determinants of mean probability of adequacy (MPA) * among pregnant women in Uttar Pradesh, India.

	Unadjusted		Not Adjusted for Energy		Energy Adjusted	
Variable	Coefficient (95% CI)	*p*-Values	Coefficient (95% CI)	*p*-Values	Coefficient (95% CI)	*p*-Values
	*N* = 627		*N* = 627		*N* = 627	
Lacto-vegetarian	0.04 (0.01, 0.07)	0.01	0.02 (−0.01, 0.05)	0.29	0.01 (−0.01, 0.03)	0.28
Food Insecure			−0.02 (−0.06, 0.02)	0.36	−0.003 (−0.03, 0.02)	0.82
Caste						
General			Reference		Reference	
SC/ST			−0.02 (−0.07, 0.03)	0.40	−0.007 (−0.04, 0.02)	0.65
OBC			−0.0004 (−0.04, 0.04)	0.98	−0.001 (−0.03, 0.03)	0.92
Maternal Education						
No Schooling			Reference		Reference	
Primary School			0.03 (−0.02, 0.08)	0.27	−0.03 (−0.06, 0.004)	0.09
Secondary and above			0.04 (0.00, 0.08)	0.05	−0.01 (−0.04, 0.01)	0.41
Parity			−0.005 (−0.01, 0.004)	0.29	−0.002 (−0.01, 0.003)	0.38
SES						
1 (Lowest Tertile)			Reference		Reference	
2			−0.005 (−0.04, 0.03)	0.80	−0.02 (−0.04, 0.003)	0.09
3 (Highest Tertile)			0.02 (−0.02, 0.07)	0.50	−0.005 (−0.03, 0.02)	0.69
Energy (kcal/day) **					0.41 (0.39, 0.44)	<0.001

MPA: mean probability of adequacy; OBC: other backward classes; SC: scheduled caste; ST: scheduled tribe. * Square-root transformed; ** natural-log transformed.

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
