# Peer review of "Nutrient Adequacy Is Low among Both Self-Declared Lacto-Vegetarian and Non-Vegetarian Pregnant Women in Uttar Pradesh"

_nutrients, 2020, doi:10.3390/nu12072126_

Round 1

Reviewer 1 Report

The article entitled " Lacto-vegetarian pregnant women in Uttar Pradesh, India have moderately better diets than non-vegetarian women because of the intersections of caste and economic status" is very-good written population study, which concern nutritional habits of pregnant women divided into two groups: vegetarians and non-vegetarians.

There is some minor changes which should be improve.

  1. Figure 3. - legend on the figure did not described all nutritional factors (i.e. readers have to guess what is the main source of vitamin b12). Both A and B part of figure 3 should have coloristic legend of all factors examined in the study.
  2. Text of article should be carefully check again for prevent grammar and style mistakes.

Author Response

Reviewer 1:

The article entitled " Lacto-vegetarian pregnant women in Uttar Pradesh, India have moderately better diets than non-vegetarian women because of the intersections of caste and economic status" is very-good written population study, which concern nutritional habits of pregnant women divided into two groups: vegetarians and non-vegetarians.

There is some minor changes which should be improve.

  1. Figure 3. - legend on the figure did not described all nutritional factors (i.e. readers have to guess what is the main source of vitamin b12). Both A and B part of figure 3 should have coloristic legend of all factors examined in the study.

Response: Thank you for pointing this. The original figure had all the nutritional factors but when reformatted some were not viewable. We have remedied this issue in the updated manuscript.

  1. Text of article should be carefully check again for prevent grammar and style mistakes.

Response: We have reread the manuscript to correct all grammar and style mistakes.

Reviewer 2 Report

Dear Editor:

Thank you for the opportunity to review the manuscript entitled “Lacto-vegetarian pregnant women in Uttar Pradesh, India have moderately better diets than non-vegetarian women because of the intersections of caste and economic status”. I read the manuscript with much enthusiasm and appreciate the contribution of the manuscript to Indian women. Additionally, it is one of the manuscripts on Indian women that have used rigorous methodology to assess the micronutrient adequacy.

My suggestions are appended below:

  1. The title could be shortened than in its current form.
  2. In India, many women report them as nonvegetarians, however, they practice vegetarianism due to varied reasons. A line could be added to the methodology of how the authors confirmed the food habit as vegetarian and non-vegetarian in these pregnant women. Was this only based on self-reporting?
  3. One of the major points that could have been considered in the manuscript was the supplement intakes during pregnancy. The pregnant women during pregnancy, especially in the second and third trimester are supplemented with multiple micronutrients these days, which adds to the micronutrient intakes. Additionally, a combination of diet and supplement intake information could have provided more information on the current scenario- to the data presented in this manuscript (as this manuscript focus on micronutrient adequacy as well).
  4. The study has taken only verbal consent from the participants. Was this approved by the IRBs were the approval was obtained? Good to add a line on this as it would protect the authors.
  5. Line 97-98: A score of five or more has been associated with an increased likelihood of nutritional adequacy [31]. The FGI based micronutrient adequacy estimates have been derived for non-pregnant and lactating women (with moderate correlation), however, the present data is from pregnant women. In the absence of such information, it would be prudent to specify the same for “non-pregnant women”.
  6. In the study population, a line on the participant gestation, i.e. trimester, would provide more clarity to the readers as this is missing.
  7. The information on unhealthy food is provided in the methodology but totally missing in the results. A few lines could be added to the results section to justify the statements regarding the same in the discussion section.
  8. The discussion section could be shorter and focused on women’s dietary/nutrient intakes.

Author Response

Reviewer 2:

Thank you for the opportunity to review the manuscript entitled “Lacto-vegetarian pregnant women in Uttar Pradesh, India have moderately better diets than non-vegetarian women because of the intersections of caste and economic status”. I read the manuscript with much enthusiasm and appreciate the contribution of the manuscript to Indian women. Additionally, it is one of the manuscripts on Indian women that have used rigorous methodology to assess the micronutrient adequacy.

My suggestions are appended below:

1. The title could be shortened than in its current form.

Response: We have shortened the title and have updated it based on feedback received from presentation of that abstract at the American Society for Nutrition conference. The title now reads “Nutrient adequacy is low among both self-declared lacto-vegetarian and non-vegetarian pregnant women in Uttar Pradesh”

 2. In India, many women report them as nonvegetarians, however, they practice vegetarianism due to varied reasons. A line could be added to the methodology of how the authors confirmed the food habit as vegetarian and non-vegetarian in these pregnant women. Was this only based on self-reporting?

Response: Our main exposure of interest was self-reported vegetarian/non vegetarian dietary preferences. We had no other way of confirming food preferences besides self report.

 3. One of the major points that could have been considered in the manuscript was the supplement intakes during pregnancy. The pregnant women during pregnancy, especially in the second and third trimester are supplemented with multiple micronutrients these days, which adds to the micronutrient intakes. Additionally, a combination of diet and supplement intake information could have provided more information on the current scenario- to the data presented in this manuscript (as this manuscript focus on micronutrient adequacy as well).

Response: Our previous study in the same population showed very low consumption of iron folic acid (IFA) and calcium supplements (Nguyen et al 2019). For example, only 8% of pregnant women consumed 100+ IFA tablets or only 2% met the recommendation of consuming at least 180 IFA tablets during pregnancy. Consumption of calcium was even lower, only 1% and 3% women consumed 180 and 100 tablets, respectively. Therefore, we expect that combination of diet and supplement intake would not change micronutrient adequacy.

Ref: Nguyen PH, Kachwaha S, Avula R, Young M, Tran LM, Ghosh S, Agrawal R, Escobar-Alegria J, Patil S, Menon P. Maternal nutrition practices in Uttar Pradesh, India: Role of key influential demand and supply factors. Matern Child Nutr 2019; 15:e12839.

 4. The study has taken only verbal consent from the participants. Was this approved by the IRBs were the approval was obtained? Good to add a line on this as it would protect the authors.

Response: Yes, Verbal informed consent was approved by ethical review board (IRB) from Suraksha Independent Ethics Committee in India IRB (approval number: 2017-10-9094) and the International Food Policy Research Institute (IRB approval number: 17-09-11).

5. Line 97-98: A score of five or more has been associated with an increased likelihood of nutritional adequacy [31]. The FGI based micronutrient adequacy estimates have been derived for non-pregnant and lactating women (with moderate correlation), however, the present data is from pregnant women. In the absence of such information, it would be prudent to specify the same for “non-pregnant women”.

Response: Thank you for this important suggestion. We edited to sentence to now say “A score of five or more has been associated with increased likelihood of nutritional adequacy in non-pregnant non-lactating women of reproductive age ”(line 99).

 6. In the study population, a line on the participant gestation, i.e. trimester, would provide more clarity to the readers as this is missing.

Response: Thank you for noting this was missing from our study population description. We have added that we selected households with a pregnant woman in their second or third trimester of pregnancy to the study population section of the methods on page 2 (lines 79-80).

 7. The information on unhealthy food is provided in the methodology but totally missing in the results. A few lines could be added to the results section to justify the statements regarding the same in the discussion section.

Response: On lines 203-206 in the results section we provide information on unhealthy food consumption in each group: “Unhealthy food consumption (eg: salty and sweet snack foods) was also higher in lacto-vegetarian (40.9%) than non-vegetarian women (30.4%), but adjusted analysis found no statistically significant difference between groups (p>0.05).”

 8. The discussion section could be shorter and focused on women’s dietary/nutrient intakes

Response: We have revised the discussion to make it more concise and focused on women’s dietary/nutrient intakes as suggested.